



# Technical note: An innovative monitoring approach to measure spatio-temporal throughfall patterns in forests

Lea Dedden[1] & Markus Weiler[1]

[1]Chair of Hydrology, Albert-Ludwigs University, Freiburg, 79098, Germany

*Correspondence to*: Lea Dedden (lea.dedden@hydrology.uni-freiburg.de)

**Abstract.** Throughfall in forests is spatially highly heterogeneous creating distinct patterns that persist over time and propagate with infiltration into the soil. Despite its importance for forest ecohydrological processes, experimentally derived high-quality datasets describing spatio-temporal throughfall dynamics at fine temporal and spatial resolution are still scarce. The majority of studies were unable to measure throughfall at high temporal and/or spatial resolution because of extensive sampling efforts, especially in forests with complex structures. We present a new, innovative throughfall monitoring approach for continuous, automated measurement of throughfall without removing the water for infiltration that allows to quantify the spatio-temporal throughfall variability at both intra-event and intra-stand levels. The network captures spatial throughfall patterns and their temporal persistence across rainfall events of varying size during leafed and non-leafed periods. The throughfall monitoring network features 60 self-built, cost effective throughfall samplers, with four throughfall collection compartments and tipping bucket units each connected to a newly developed microcontroller board enabling fully automated, low-maintenance operation during rainfall events. The network, collecting data since the winter of 2024/2025, is setup in a stratified sampling pattern among four forest plots of Beech, Douglas fir, Silver fir, and mixed trees in a mature temperate forest in Germany. Throughfall data from a four-week observation period in the spring of 2025 are included in this study to showcase the potential of this approach. The data support the networks' ability to capture small-range spatio-temporal throughfall patterns across the study area.



## 1. Introduction

Rainfall reaching the tree canopy of a forest partitions into interception, stemflow and throughfall. Throughfall (TF) is the largest water input fraction in most forest ecosystems exhibiting high spatial heterogeneity due to redistribution within the tree canopy (Levia et al., 2011). The emerging spatio-temporal patterns may persist over time and determine the variable throughfall input to the forest floor which is depleted at some points (Holwerda et al., 2006; Keim et al., 2006) and exceeds even open rainfall at other points (Lloyd and Marques F., 1988; Siegert et al., 2016).

The main controls for the spatio-temporal variability of throughfall are vegetation structure and composition (Staelens et al., 2006b; Zimmermann et al., 2008a), topography (Siegert et al., 2016) and regional meteorology including precipitation (Levia and Frost, 2006; Raat et al., 2002; Siegert et al., 2016). The general mass balance is (adapted from Carlyle-Moses (2004)):

$$TF = P_g - (I_c + SF) \tag{1}$$

with $TF$ = throughfall; $P_g$ = gross precipitation; $I_c$ = canopy interception; $SF$ = stemflow (all in mm). The spatially 35 heterogeneous character of throughfall influences the forest water balance - in particular near-surface hydrological processes - and potentially propagates into the soil (Fischer et al., 2023; Schume et al., 2003; Shachnovich et al., 2008; Zimmermann et al., 2009). In addition, throughfall functions as a nutrient pathway from the canopy to the ground (Zimmermann et al., 2008a) and is linked to biogeochemical processes and plant-water availability near the forest floor (Dalsgaard, 2007; Raat et al., 2002).

### 1.1. Experimental throughfall measurement approaches: potentials and limitations

The crucial role of throughfall as the major water flux from the forest canopy to the forest floor shown in Figure 1 has motivated a broad range of experimental studies dedicated to the challenging task of accurately estimating throughfall volumes and intensities, solute inputs and the spatio-temporal variability of throughfall (Keim et al., 2005; Levia and Frost, 2003, 2006; Link et al., 2004; Pypker et al., 2005). The last three decades have seen a rise in interest as research has begun to examine spatio-temporal throughfall patterns across different ecosystems, vegetation structures, and spatial scales, as well as the 45 intricate relationship to biotic and abiotic controls (Bialkowski and Buttle, 2015; Levia et al., 2019; Staelens et al., 2006b). The principle measurement approach of most studies characterize three components: the collector type, the sampling size and the sampling design (Zimmermann and Zimmermann, 2014). Throughfall samplers are either individual funnels ( approx. 500 cm² orifice) or troughs (several meter long and several square meter orifice; both see Figure 1) arranged in a stratified or random design. Sample sizes range from tens to hundreds of samplers supplemented by a nearby rain gauge for gross 50 precipitation measurements. Readings are either continuous or at an event basis or a larger interval.

#### a. Sample size

Substantial effort has been made to obtain the most accurate throughfall estimates (Carlyle-Moses et al., 2014; Crockford and Richardson, 2000; Thimonier, 1998). The frequently used method of Kimmins (1973) uses the coefficient of variation to calculate the number of required collectors (sample size) to measure representative throughfall averages for a given confidence 55 interval and precision (at pre-set small mean TF errors) (Carlyle-Moses, 2004; Lloyd and Marques F., 1988; Rodrigo and





Àvila, 2001). The required sample sizes are often large and not feasible for experimental setups. The deployment of roving samplers - contrary to samplers at fixed locations - increases the number of sampling locations through relocation (Kimmins, 1973; Link et al., 2004; Lloyd and Marques F., 1988; Rodrigo and Àvila, 2001; Ziegler et al., 2009) and thereby reduces the sample size, but can only be used to estimate long-term averages.

**b. Sampler type**

The quality of the throughfall measurement is determined by the type of collector (trough or funnel) in addition to the sample size. The collector efficiency depends on spatial throughfall structure: funnel sampler generally offer high spatial resolution (Zimmermann et al., 2010) and may capture very short-range variation (few cm) or throughfall dripping edges. Troughs are quite efficient for small sample sizes reducing random errors, integrating outliers and covering large areas below the canopy
(Zimmermann et al., 2010; Zimmermann and Zimmermann, 2014). Often, funnel and trough collectors are combined.

**c. Sampling design**

According to Thimonier (1998) and Zimmermann et al. (2010), the sampling strategy or design is more important than the sample size or type of sampler. For example a random distribution can avoid clustering, assure representative sampling and support geostatistical analysis approaches (Metzger et al., 2017; Zimmermann et al., 2009). In terms of measurement
frequency, automated or manual readings define the temporal and spatial resolution of the data and consequently its scope of application. The absence of a consensus on reference standard devices and schemes for throughfall measurements is emphasized by the substantial body of research with varying experimental and analytical approaches (Levia and Frost, 2006; Llorens and Domingo, 2007; Zimmermann and Zimmermann, 2014). Different choices of spatial scales, measurement setups and ecosystems under study limit the comparability and transferability of study findings (Crockford and Richardson, 2000;
Lloyd and Marques F., 1988). Zimmermann et al. (2016) showed that existing measurement schemes are frequently not optimally matched to the system under study, omitting relevant factors such as the study area extent. Mismatches between required theory and practical feasibility (Thimonier, 1998; Zimmermann and Zimmermann, 2014) can produce large errors in the data (Kimmins, 1973; Thimonier, 1998; Zimmermann et al., 2010).

Throughfall is most variable at small scales (Wullaert et al., 2009) and varies as function of canopy complexity, tree density
and precipitation magnitude (Rodrigo and Àvila, 2001; Staelens et al., 2006a). Measurements at the event or even intra-event scale and with a large sample size are recommended in order to investigate throughfall variability (Staelens et al., 2006b) and temporal stability of throughfall patterns (Fischer et al., 2023; Keim et al., 2005; Staelens et al., 2006a; Zimmermann et al., 2010). However, individual rain events are rarely observed at high temporal resolution (Staelens et al., 2008) and spatio-temporal analysis of throughfall typically rely on measurement schemes of a low number of events (Raat et al., 2002; Staelens
et al., 2006a) typically missing intra-event variability.



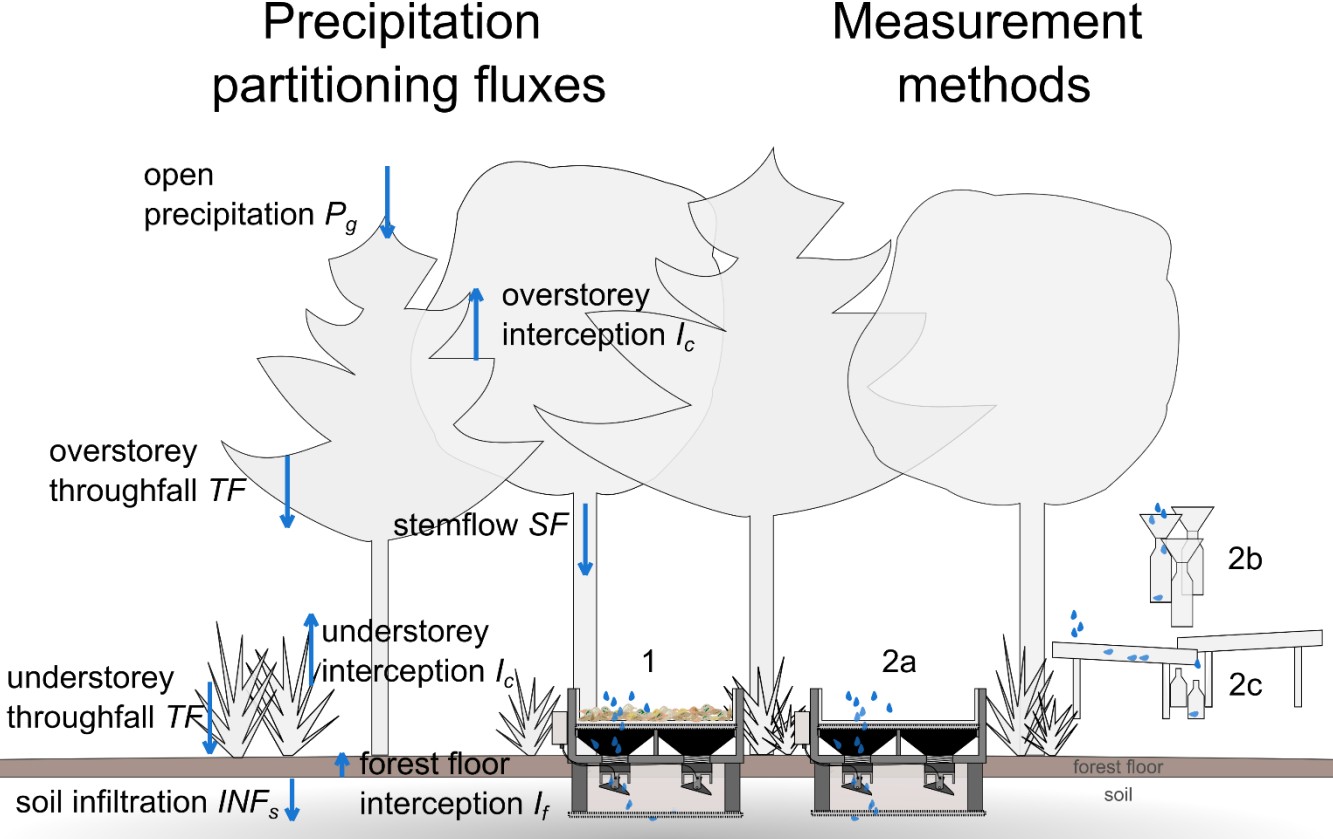

**Figure 1: Schematic overview of precipitation partitioning in a forest (left) and throughfall measurement methods (right). (1) shows the throughfall sampler with forest floor inlay measuring soil infiltration, (2a-c) show throughfall measurements: the throughfall sampler without forest floor inlay (2a), funnel (2b) and trough (2c) collectors.**

### 1.2. Experimental throughfall measurement approaches: new directions

To address the outlined challenges, high quality throughfall data are needed, collected with appropriate measurement designs and devices (Germer et al., 2006; Levia and Frost, 2006; Zimmermann et al., 2009; Zimmermann and Zimmermann, 2014). In order to observe small-scale variability at the individual tree level and to representatively cover entire stand vegetation structures at high spatial resolution, these schemes should ideally have large number of samplers with adequately sized samplers at fixed locations (Fischer et al., 2023; Metzger et al., 2017; Zimmermann et al., 2010). Only measurements at high temporal resolution on intra-event scale allow to investigate canopy storage, interception changes and temporal stability of spatial throughfall patterns (Keim et al., 2005; Zimmermann and Zimmermann, 2014). Ideally, these schemes integrate automated and continuous measurement devices that collect long-term high quality data which meet the requirements for the analysis of seasonality effects (Zimmermann et al., 2008b) and long-term changes of throughfall spatio-temporal patterns (Link et al., 2004). Together, these schemes would then contribute to a deeper understanding of throughfall controls like rainfall



regimes, relationship to near-surface hydrological processes and canopy interception variability in forest stands (Blume et al., 2022; Zimmermann et al., 2008a; Zimmermann et al., 2008b). Furthermore, it might improve water flux modelling of watersheds providing better insights into water losses and yields of forested areas (Crockford and Richardson, 2000; Zimmermann et al., 2008b).

We present a novel throughfall measurement network of ground-level tipping bucket samplers with 240 individual collection compartments for quantification of throughfall spatio-temporal variability on an intra-event and intra-stand scale. The systems were set up in forest plots of pure Beech, pure Douglas fir, pure Silver fir and a mixed Beech Douglas fir plot. The automated, continuous measurement produces long-term time series at a high spatio-temporal resolution with moderate maintenance effort that can be used to investigate temporal persistence, long-term changes in throughfall, and spatial species-specific throughfall

patterns. In addition, it causes minimal disruption to near-surface hydrological processes.

In order to gather data on throughfall spatio-temporal variability, the study aims are to (i) design a throughfall sampler that allows continuous, automated and minimally invasive data collection, (ii) develop a sampling scheme using a combination of samplers that accurately measures throughfall variability across plots of different tree compositions; and (iii) implement the sampling scheme at the study site of a temperate, mature forest stand.

## 115   2.   Study area and methods

### 2.1. Throughfall measurement sampler

#### 2.1.1.    Sampler design

The throughfall sampler depicted in Figure 2 uses a rectangular, solid plastic container (60 x 40 x 20 cm, Euro container) divided into four equal trough drainage compartments. Each compartment, with a 600 cm$^2$ collection area, collects and funnels

throughfall water to a separate tipping bucket unit for water flux measurement. The container's inner walls are lined with water and UV-resistant pond liner, clamped by aluminium ledgers to create sufficient slope for water drainage. Additionally, the aluminium ledgers serve as mounts for a detachable metal grid and geotextile upon which forest floor litter can be placed or will fall naturally (Figure 2). Hence, the metal grid in combination with the geotextile filters the throughfall to prevent clogging of the tipping bucket. The pond liner is attached to an opening at the bottom exit of the container with a commercial sink drain

in each compartment. A nylon mesh filter keeps small particles out of the drain. The tipping bucket units are fastened to the drain treads underneath the container, allowing them to tip freely into the open area below (Figure 2). The whole throughfall containers are levelled on porous concrete stones (25 x 25 x 20 cm) with a hollow centre. Prior to installation, topsoil of 30 cm depth was removed at each sampler location and a concrete stone together with a metal grid were inserted. This setup creates a flat, sturdy mounting that prevents tipping of the container and allows infiltration of the quantified water fluxes into

the soil. Unlike the typical large troughs, this design is minimally invasive to near-surface hydrological processes and tree water supply. It also enables direct measurement of other processes in the soil like soil water content. In addition, the water





could also be sampled underneath for water quality measurements. The stone base also makes maintenance easier, like exchanging tipping buckets, and prevents animal intrusion. Positioning the containers at ground level minimizes measurement inaccuracies caused by wind generating typical under-catch of rainfall gauges.


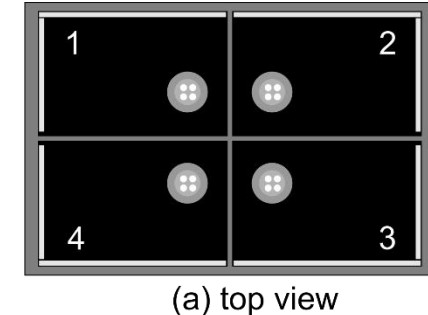

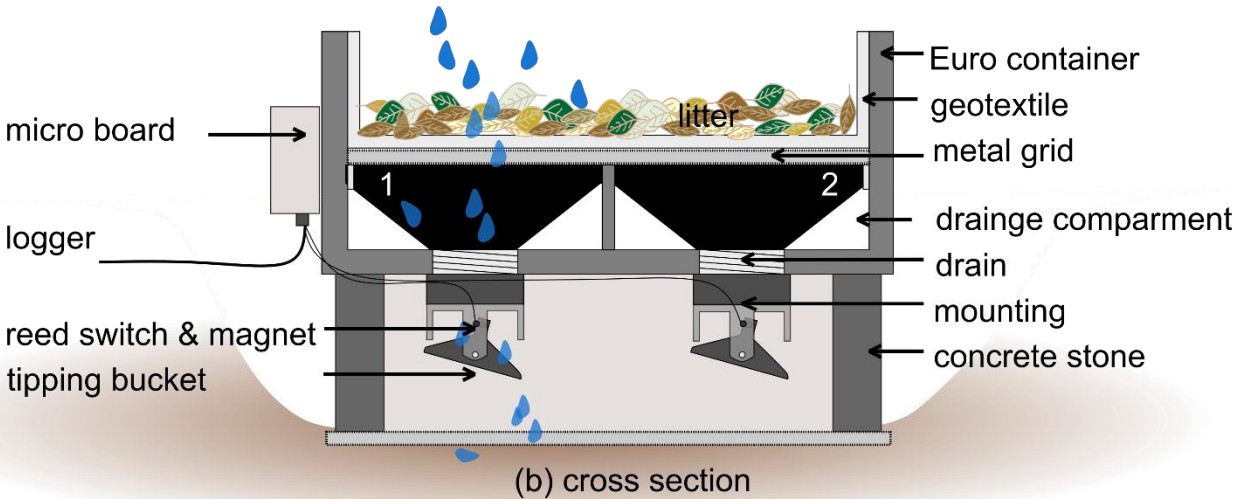

**Figure 2: Design of throughfall sampler with four equal drainage compartments (top view), each equipped with single tipping bucket units for measurement (cross section).**

### 2.1.2. Forest floor inlay

The effect of forest floor interception (Gerrits et al., 2007; Paulsen and Weiler, 2025) that reduces throughfall to the fraction infiltrating into the mineral soil, can be measured by adding unfragmented forest floor litter to the throughfall sampler (Figure 2 & Figure 2). In case this is not of interest, the throughfall sampler can be deployed without forest floor litter directly obtaining throughfall data above the forest floor (see (2a) in Figure 1). For our study, we decided to include the interception of the forest floor in our measurements (Gerrits et al., 2007; Paulsen and Weiler, 2025) and, to directly investigate the propagation of the

overall throughfall patterns into near-surface soil moisture (Fischer et al., 2023; Metzger et al., 2017; Raat et al., 2002; Schume et al., 2003). Adding litter into the samplers has the practical benefit of lowering measurement errors since the organic material





prevents splash out of the sampler. As for this study, samplers with a forest floor addition were used, the throughfall samplers are referred to as infiltration samplers thereafter.

### 2.1.3. Tipping bucket units

The 3D-printed tipping buckets consist of two parts: a mounting with a reed switch and threaded top, and a tipping bucket with a magnet on top (Figure 2). The tipping bucket is attached to the mounting by a stainless-steel pin. Both components are made from durable and chemically resistant PETG, with the inner surfaces of the tipping bucket units ironed reducing water adhesion and rewetting effects. For further information on the design and implementation of the tipping bucket unit see Paulsen and Weiler (2025). A 4 mm diameter pipe inside the mounting channels water from the drain to the tipping bucket, guaranteeing a

maximum flow rate that cannot be exceeded. During very high intensity rainfalls, the drainage compartments can store and buffer up to 1 L of excessive water. Tipping buckets with a defined volume of 2.7 mL alternately fill up with water, tip to empty and thereby move the magnet past the reed switch. This mechanism triggers the reed switch to sense out short electric pulses. The pulses are recorded by the micro board as tip counts per defined time interval. Counted tips are converted to water volumes and divided by collection area of the infiltration sampler compartments to calculate the infiltration depth (mm) for

every compartment.

The micro boards were designed as a cost-effective solution for continuous, automated measurement of infiltration water volumes. They each accommodate 4 pulse count inputs and an extension for four additional pulse count inputs. A micro controller (AVR SAMD21) is used to realize the pulse inputs as interrupts, guaranteeing that no pulses are missed. Two steps are involved in debouncing the tipping bucket units: each reed switch has a resistance capacitor filter linked to reduce bouncing.

Laboratory tests with debouncing tipping bucket units produced 500–800 ms as ideal timeout settings preventing erroneous double tips" while still capturing all pulses for tested maximum rainfall intensity of less than 90 mm h$^{-1}$. The micro board is connected to a Campbell Scientific CR350 data logger via SDI-12 data communication, logging at 15 min intervals. Any other SDI-12 logger could be used as well. A mini-USB port offers direct connection to a computer. For detailed descriptions on the micro board hardware and programming please refer to Paulsen and Weiler (2025).

### 170 2.2. Site description

The infiltration samplers are part of an Ecohydrological sensor network located at the ECOSENSE forest research site (Figure 3) near Ettenheim (48.2685° N, 7.8782° E), between the Upper Rhine Valley and the Black Forest in south western Germany. The elevation of the research site ranges from 480 to 500 m a.s.l., the climate is temperate-humid (Köppen *Cfb*) with mean annual precipitation of 1120 mm and mean annual temperature 9.5° C in reference period 1961–1990 (Landesanstalt für

Umwelt Baden-Württemberg, 2024). Occasional summer drought can occur (year 2003 & 2018). The northern part of the research site is located on Cambisols over Triassic Buntsandstein, while the southern part features Pseudogley soils over Triassic Muschelkalk (Werner et al., 2024). The forest floor averages 10 cm in thickness and has been classified as moder (Landesamt für Geologie, Rohstoffe und Bergbau Baden-Württemberg, 2024).



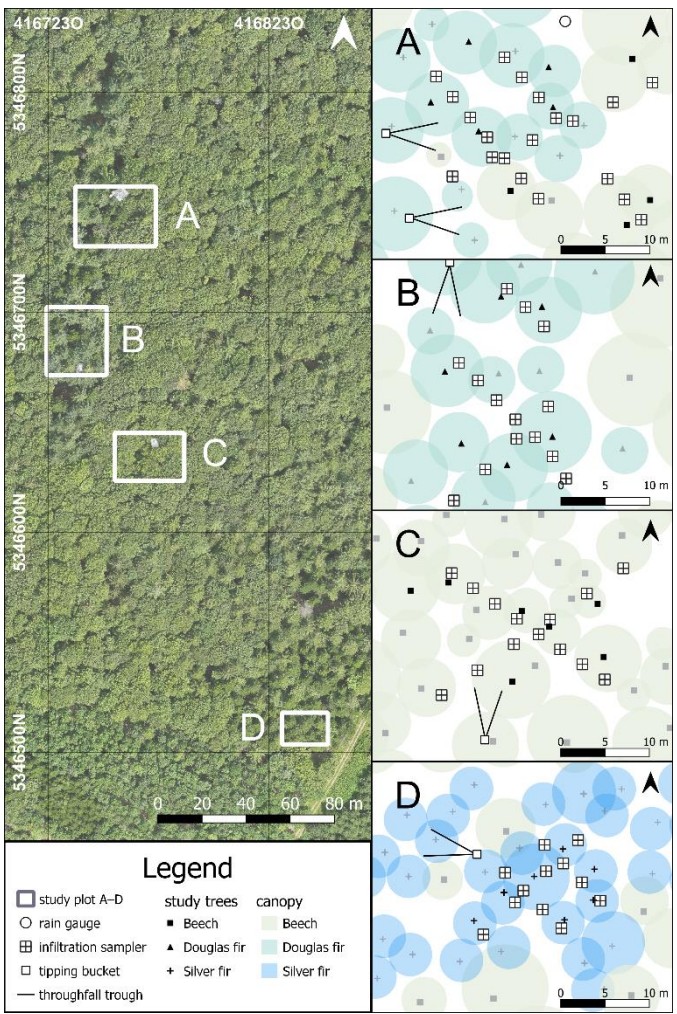

Figure 3: ECOSENSE forest research site near Ettenheim, Germany. To the right the infiltration measurement network of each study plot A–D with mixed Beech and Douglas fir (A), pure Douglas fir (B), pure Beech (C) and pure Silver fir (D).

The research site is covered by a European Beech-dominated mixed forest managed for timber production. In addition to Beech trees (*Fagus sylvatica*), the stand is composed by patches of Silver fir (*Abies alba)* and Douglas fir (*Pseudotsuga menziesii*), aged approximately 80, 60 and 50 years, respectively (Landesamt für Geoinformation und Landentwicklung Baden-Württemberg, 2025). Individual spruce and oak trees (Werner et al., 2024) complement the average tree density of 500 trees ha$^{-1}$. With the exception of a few younger 10–20 m tall Beech and Silver fir trees, there is no understorey vegetation, and the overstorey reaches heights of 25–35 m. The study area shown in Figure 3 was divided into four subplots: a mixed Beech & Douglas fir plot (A), pure Douglas fir (B) and pure Beech (C) in direct vicinity on a plateau and a Silver fir plot (D) located 200 m southeast on a gentle southeastern slope (Figure 3). Each subplot is approx. 400 m$^2$ large and includes 7–10 selected "measurement trees" of comparable age and height.





### 2.3. Throughfall measurement network

The throughfall measurement network of 60 infiltration samplers is installed in a stratified design of perpendicular transects across the four plots (Figure 3 A–D). This distribution supports measurements below various canopy positions, including stem closeness, canopy centre, and canopy edge, and prevents clustering while guaranteeing coverage of the variance of the forest
structure (Holwerda et al., 2006; Link et al., 2004). The network design combines a model-based and a design-based component as suggested by Zimmermann and Zimmermann (2014) to account for objective assessment of uncertainty together with purposive measurement such as for interest in distribution across the entire area. The sampler design of four neighbouring trough and tipping bucket units adds measurement points in direct vicinity (model-based component). This supports outlier detection and is favourable for geostatistical analysis approaches improving variogram estimation (Voss et al., 2016;
Zimmermann et al., 2009). The distribution of the samplers on stratified transects with 3 and 5 m spacing in NW-SE and NE-SW orientation (design-based) ensures an unbiased estimate of spatial mean throughfall. Furthermore, variably spaced transects also support the variogram development and enable the estimation of short- and long range throughfall variability including anisotropy (Schume et al., 2003). Beginning in December 2024, continuous infiltration measurements of the first installed infiltration samplers were conducted. Regular maintenance has been performed on all samplers.

Figure 3 also depicts the five throughfall trough collectors, which are positioned at random throughout each study plot, and two rain gauges for gross precipitation complement the setup. The trough collectors consist of two V-shaped stainless-steel troughs of 0.3 m² receiving area and a length of 3 m installed 1 m above ground at an 45° angle relative to each other. The troughs channel collected water into a tipping bucket rain gauge (Rain collector II 7852, Davis Instruments, resolution 0.2 mm) covered by a nylon mesh. Gross precipitation is measured with tipping bucket rain gauges (YOUNG Tipping bucket rain
gauge 52202, resolution 0.1 mm). They were placed at on opening located at 200 m distance from plots mounted 1 m above ground and above the canopy at 47 m height installed on a measurement tower located at the centre of the mixed plot (Figure 3). Precipitation data were wind-corrected according to Kochendorfer et al. (2017).

### 3. Results

#### 3.1. Calibration of the tipping bucket unit

Every individual tipping bucket unit was calibrated in the laboratory to determine single tip volumes. 3D-printed inflow regulation plugs were screwed onto PET laboratory bottles of 100 mL and plugged upside down into the drains of each compartment dripping at constant rate of around 1 mL s⁻¹ (Bialkowski and Buttle, 2015; Levia et al., 2019; Staelens et al., 2006b). For all tipping buckets the average tipping bucket volume is 2.87 mL (SD = ±0.28) resulting in a resolution of 0.05 mm tip⁻¹ and an accuracy of 6.4 % calculated as mean percent error of measurements based on actual tip volume (2.7 mL).
The tip resolution and accuracy are comparable with other commercial tipping bucket rain gauges (e.g. HOBO 0.2 mm resolution with 1% accuracy; YOUNG 0.1 mm resolution with 2% accuracy; DAVIS 0.2 mm with 4% accuracy). Four different



3D printers (Bamboo Lab P1P, Original Prusa i3 MK2, Raise3D Pro2, Ultimaker4) were used to manufacture the 240 tipping buckets in order to speed up the manufacturing process. The calibration results in Figure 4 show differences in the average tip volumes ranging between 2.21 to 3.69 mL and average tipping bucket weights between 9.3 to 11.1 g across the print batches,

despite the use of identical CAD input files. These variations start with varying printer accuracies and subsequently propagate into the tipping bucket volume measurements. Because of undercatch effects, the average tip volume from calibration decreases with the weight of the tipping bucket. Distinct boxplot IQRs of 0.2 mL (black) to 0.5 mL (transparent) (Figure 4) show the variability of within-batch precision of the individual printers. The calibration results highlight the importance of individual calibration coefficients for each tipping bucket unit and the need for regular field calibration to assure continuous high data

quality. Infiltration volumes and according depths were derived from tip counts of tipping bucket units as following:

$$INF_{depth} = \frac{(x_{pulse} \times c_{calibration})}{A_{collection}} \tag{2}$$

with $x_{pulse}$ = counted tips, the individual calibration coefficient $c_{calibration}$ = 2.21 to 3.69 mL and a collection area of $A_{collection}$ = 600 cm$^2$.

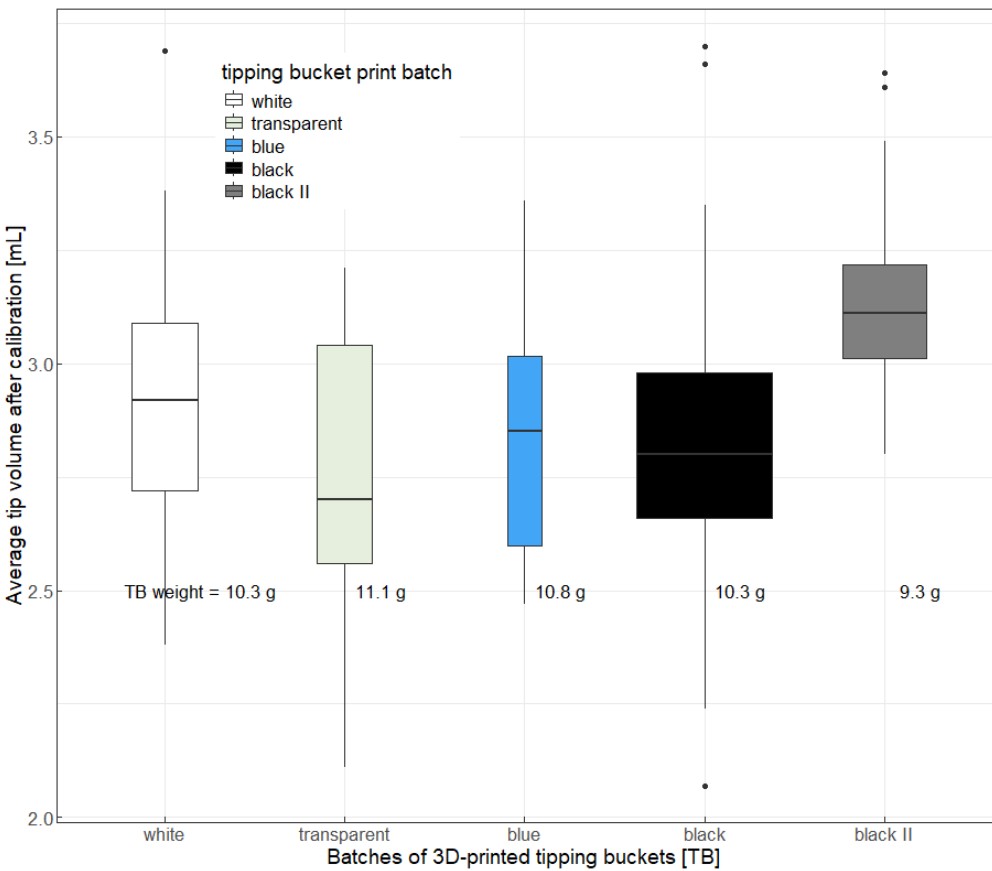

**Figure 4: Calibration of 240 tipping bucket units with boxplots showing average tip volumes of tipping bucket units from five 3D-printing batches (batches defined by filament colours white, transparent, blue, black I & II).**




### 3.2. Spatio-temporal infiltration dynamics from infiltration network data

#### 3.2.1.    Separation of events and development of event infiltration coefficient

Continuous in situ infiltration data from all four subplots were available starting April 2$^{nd}$, 2025. The corresponding time series
of wind-corrected (Kochendorfer et al., 2017) hourly gross precipitation ($P_g$) from the tower station was separated for
individual rainfall events and validated against ground station precipitation data. Events were defined as periods of continuous
hourly $P_g \geq 0.1$ mm with a maximum of one hour $P_g \leq 0.1$ mm in between and a minimal total event $P_g \geq 2.5$ mm. An event
ends when $P_g \leq 0.1$ mm for two consecutive hours or more. When applied to the infiltration data, the event separation accounts
for canopy drip from open throughfall and forest floor percolation up to two hours post-event. The observed data showed that
this timeframe is well suited as most infiltration events ended 2 hours after precipitation ceased. 'Last drips' from infiltration
samplers recorded hours past the event were not included. Infiltration amounts from these periods were marginal and can be
neglected.

Based on the separated precipitation events infiltration coefficients ($C_{inf}$) - following the principle of the runoff coefficient e.g.
Blume et al. (2007) or Savenije (1996) - were developed for every tipping bucket unit and event. Event $C_{inf}$ of a throughfall
bucket unit describes the total event infiltration at a location as a fraction of the total event gross precipitation and thereby
facilitate the comparison of event infiltration across the four plots. It indicates how $P_g$ is redistributed by forest vegetation: the
percentage of $P_g$ lost due to interception and the percentage of $P_g$ infiltrating into soil.

#### 3.2.2.    Infiltration variability of an example event

The selected example of a moderate precipitation event occurring April 17$^{th}$ 2025 (total event $P_g$ = 11.2 mm, 28 h, max.
intensity of 1 mm h$^{-1}$) presented in Figure 5 illustrates the spatio-temporal variability of throughfall and resulting event
infiltration for the four forest stands Beech, Douglas fir, Silver fir and mixed trees. The median total event infiltration is highest
at the Beech plot with $INF_{median}$ = 9.7 mm followed by the Douglas fir and mixed plot ($INF_{median}$ = 9.4 and 9.2 mm) whereas
the Silver fir plot has the lowest median infiltration with $INF_{median}$ = 6.9 mm.

Throughout the event, the median infiltration and all single sampling locations follow the overall trend of gross precipitation
(Figure 5). Early-stage infiltration is lower than precipitation (approx. 50% $P_g$) and indicates filling up of canopy and forest
floor water storage. The cumulative infiltration curves of all recording tipping bucket units show a higher spatial
heterogeneity in infiltration depth across the Beech and Douglas fir plot, then at the Silver fir and mixed plot. At several
sampling locations across all plots infiltration during the event is low but steady with consistent hourly infiltration < 0.3 mm
h$^{-1}$ and an event infiltration < 6 mm. The increasing precipitation intensity on April 17$^{th}$ 2025 at 02:00 h to 07:00 h LT
translates directly into increasing infiltration.







**Figure 5: Spatio-temporal infiltration variability during a precipitation event on April 17th 2025 (total event Pg = 11.2 mm, 28h) measured by infiltration samplers at four subplots Mixed (A), Douglas fir (B), Beech (C) and Silver fir (D) at the ECOSENSE field site**

Intra-stand spatio-temporal variability of event infiltration is visualized by heat maps of all tipping bucket units recording the selected event (Figure 5 lower panels): General spatio-temporal infiltration patterns resemble among plots with infiltration ranging from 0 to 3 mm h$^{-1}$ across measurement points of few meters' distance. Precipitation periods of $P_g \geq 0.75$ mm h$^{-1}$ (April 17$^{th}$ 2025, 02:00 to 07:00 h & 17:00 h LT) generate peak infiltration ($\geq 3$ mm h$^{-1}$) at some locations in the Silver, Douglas fir- and Beech plot, however the majority of tipping bucket units in all four plots record infiltration up to 1 mm h$^{-1}$. In particular, the first 2 and the last 4 hours of the event characterize low infiltration depths in line with gross precipitation depths.





### 3.2.3. Infiltration variability of observation period

Figure 6 compares the selected events (total event $P_g \geq 2.5$ mm) that occurred during a two-week period end of April 2025, in which foliage has been already well developed. The total event precipitation range between 3.2 and 11.2 mm, with durations between 4 to 28 h. Comparing the events, the median infiltration coefficients increase with total event precipitation for the Douglas fir, Mixed and less pronounced for the Silver fir plot. Infiltration coefficients of the Beech plot show no distinct relationship to event precipitation magnitude. Events of duration $\geq 10$ h (event 2 & 6) show larger variability in infiltration across the single plots than events of duration $\leq 10$ h (event 1, 3, 4, 5). This larger heterogeneity is visible at the boxplot IQRs of 0.9 to 1 for $C_{inf}$ (event 2, Beech & Douglas fir plot) together with several locations with a $C_{inf}$ up to 2.5 (scatter) and a large number of locations with low or no infiltration measured.

The influence of event duration on spatio-temporal variability becomes more evident comparing event 1 & 6. Both events of approx. $P_g = 7$ mm differ in duration (4 to 11 h). The median infiltration coefficients are similar for all species groups for the two events, however the longer event 6 characterizes a stronger infiltration variability shown by the large IQR of Beech, Douglas fir and mixed boxplot quartiles expanding from 0 to 0.9 $C_{inf}$. In contrast, the shorter event 1 appear to produce spatially a comparably homogenous infiltration (comparably narrow IQRs of < 0.5 $C_{inf}$ with shorter upper whiskers and few locations of zero total event infiltration) due to higher mean event precipitation intensities causing fast canopy saturation and limited interception loss.

## 4. Discussion

### 4.1. Performance of infiltration measurement sampler

The throughfall samplers measured infiltration for all events of the observed period (Figure 6) demonstrating a consistent performance. The samplers capture the spatial small-scale infiltration heterogeneity within the plots as the event infiltration data in Figure 6 show: event infiltration coefficients range from 0 to 1 $C_{inf}$ across all four plots. The distinct patterns of depleted or augmented infiltration positions across the plots become more evident observing single sampler positions in the heat maps of Figure 5. Infiltration differences among neighbouring collection compartments highlight the small-scale variability and validate the sampler (compartment) support.

The cost-effective design allowed the distribution of a large sampler size (60 samplers) (Figure 3 right-side panels) with a total collection area of 11.4 m$^2$, which aligns with some of the large-scale trough systems (Germer et al., 2006; Mużyło et al., 2012; van Stan et al., 2017). In contrast, the spatial resolution of the infiltration measurement system (collection area of 600 cm$^2$) compares to sampling approaches of short-range spatial throughfall variability e.g. Carlyle-Moses (2004), Gerrits et al. (2010), Macinnis-Ng et al. (2012). This way, the samplers cover the entire stand variation in a monitoring network across the four forest plots (Figure 3 right-side panels) while retaining high spatial resolution and avoiding to integrate short-range variability with a large collection unit.





**Figure 6: Event infiltration measured by the infiltration sampler network during two weeks in spring 2025; boxplots show total event infiltration at the four subplots in relation to event Pg for all events of total Pg > 2.5 mm**

The example event (Figure 5) highlights the samplers' ability to depict the temporal infiltration variability: the continuous measurements of hourly infiltration reflect the rain intensity profile of the event with partly lagged, attenuated or depleted peaks (Dunkerley, 2015). Sampling schemes with automated, continuous readings of throughfall are rare and mostly limited in sampler size ( n < 50) (Mużyło et al., 2012; Staelens et al., 2008; van Stan et al., 2017) making them unsuited for investigating spatial throughfall characteristics. After the first months of operation, automated sampling of 240 tipping bucket units proved




to considerably reduce the sampling effort compared to manual sampling approaches of similar sampler sizes (Fischer et al., 2023; Metzger et al., 2017; Zimmermann et al., 2009). The samplers are insensible to wind and the measured water infiltrating

into the soil enables nearby soil moisture measurements. The reduced logistical and financial costs enables monitoring beyond typical timeframes of several weeks (Keim et al., 2006; Klos et al., 2014; Raat et al., 2002), a vegetation period (Cisneros Vaca et al., 2018; Molina et al., 2019; Su et al., 2019) or one to two years (Holwerda et al., 2006; Mużyło et al., 2012; Siegert et al., 2019).

Infiltration data from the observed period (3.2.3 & Figure 6) reveal that several tipping bucket units recorded none or low

infiltration for some events, while other tipping bucket units measured event infiltration up to 250% of $P_g$ (Figure 6). These measurements may reflect strong spatial heterogeneity of throughfall, interception and infiltration as a result of precipitation redistribution and funnelling within canopy and forest floor (Carlyle-Moses and Lishman, 2015; Holwerda et al., 2006; Zimmermann et al., 2009). Records of zero infiltration may inform about a blocked tipping bucket unit, but may also be a valid, valuable information of highly depleted or even absent infiltration location (Levia et al., 2019). Likewise, highly elevated

infiltration might correspond to a dripping edge position but could also arise from technical issues e.g. an unlevelled sampler or the displacement of the forest floor inlay (result is direct throughfall measurement without forest floor interception). The field application and manual construction of the samplers introduce potential sources of error e.g. water leakage from the tipping bucket units or material failure. All potential measurement errors can only be resolved with regular maintenance and calibration that was carried out to mitigate these issues, in particular before and after large events or during periods of freeze,

snow or heat stress. In addition, the collected data undergo a quality control which includes also precipitation, throughfall and air temperature data and flags suspicious measurements (e.g. infiltration without precipitation or snow melt). As the dataset of collected events expands, we can improve the distinction between valuable information and measurement errors.

Monitoring small to large precipitation events will contribute to understanding the controls of throughfall and infiltration generation at different positions (e.g. species-specific thresholds of canopy and forest floor saturation). In particular the

sampling of small precipitation events (Figure 6) remains an experimental challenge due to high spatial heterogeneity emerging from interception processes and generally small water amounts. Small events require large sampling sizes (Levia and Frost, 2003; Price and Carlyle-Moses, 2003; Rodrigo and Àvila, 2001), prior studies suggested > 200 samplers to measure throughfall from small precipitation events (Zimmermann et al., 2010). This study includes events of total $P_g$ > 2.5 mm (see Figure 6), however several smaller events – mostly under pre-wetted conditions - were recorded by a substantial fraction of tipping bucket

units demonstrating the networks' capability to sample events of $P_g$ < 2.5 mm. Precipitation characteristics control throughfall and consequently infiltration: In comparison of event 1 & 6 in Figure 6, event duration, depth and intensity appear to influence infiltration. While larger events ( > 5 mm) are generally associated with large throughfall volumes (and consequently infiltration) and reduced spatial variability (Raat et al., 2002), small events ( < 5 mm) were found to be more affected by vegetation parameters such as canopy storage capacity (Gerrits et al., 2010). The largest sampled event (Figure 6 event 2)

shows infiltration during long periods of low precipitation intensity, potentially influenced by interception e.g. "within event" evaporation (Crockford and Richardson, 1990) reflecting the uncertain role of precipitation intensity (Raat et al., 2002).



### 4.2. Further improvements for the infiltration measurement sampler

Tipping bucket rain gauges implemented in this setup (2.1.1 & 3.1) are widely used for quantifying precipitation, throughfall or stemflow due to their simple principle of operation and cost-effectiveness. However, they suffer from systematic non-linear
measurement errors depending on precipitation intensity (Marsalek, 1981; World Meteorological Organization, 2021). In particular measurements of low and high intensities are prone to errors. A correction function will be considered to address this issue for the infiltration measurement sampler e.g. Colli et al. (2013), Shimizu et al. (2018) and Stagnaro et al. (2016). As described for the example event in 3.2.2 the infiltration trend during the event compared to the precipitation profile indicate forest interception processes in form of canopy and forest floor water storage filling in the initial event phase. However, the
samplers' potential to monitor canopy and forest floor interception in detail is limited e.g. to comparisons of data from leafed and non-leafed periods or dry and pre-wetted canopy conditions. Following the steps of this research include the operation of the samplers without forest floor inlays to collect direct canopy throughfall (see 2.1.1 & 2.1.2) and a combined analysis of infiltration data with available data of lysimeters, stemflow and throughfall of the same plots. Together, this will enable a distinctive analysis of spatio-temporal throughfall patterns in relation to forest floor percolation. In order to further investigate
spatio-temporal infiltration heterogeneity, the presented infiltration measurement samplers (2.1.1) can be extended to fully equipped mini-lysimeter (Gerrits et al., 2007; Paulsen and Weiler, 2025). To achieve this, every sampler requires two load cells positioned beneath the short sides of the container. Micro boards and software can be updated accordingly, see Paulsen and Weiler (2025).

### 5.   Conclusion

The presented throughfall monitoring network at the ECOSENSE forest research site enables automated, continuous measurements of spatio-temporal throughfall dynamics across subplots of pure Beech, pure Douglas fir, pure Silver fir and mixed trees. Comprising 60 samplers with a total of 240 collection compartments and tipping bucket units, the network achieves a high spatial resolution at tree-scale while capturing the full range of throughfall variability across the plots through a stratified sampling design. The presented example data from an observation period in spring 2025 demonstrate the system
measures continuously events of all magnitudes with sensitivity to short-range spatial throughfall heterogeneity and temporal trends during events. The network complements conventional throughfall and stemflow measurements. With its automated, minimally invasive operation at low maintenance it considerably reduces the sampling effort compared to manual sampling methods making it well-suited for throughfall monitoring over several years. The growing dataset of high-resolution throughfall measurements from plots of coniferous and deciduous trees will enable the detailed analysis of species-specific
throughfall dynamics, the within-plot variability and temporal stability of patterns. It will also provide the basis for investigating how spatial throughfall patterns propagate through percolation into soil water content patterns. Data from longer operation periods will furthermore support the investigation of seasonality effects and forest age-related changes (increasing



canopy complexity) on throughfall and infiltration. Ultimately, the network and generated data contribute to our understanding of precipitation partitioning in forests especially the role of vegetation structure shaping throughfall and interception processes.


*Code/Data availability.* Code will be available on Github, data are available upon request.

*Author contributions.* LD and MW designed the infiltration measurement setup. LD conducted the field set-up and measurements with support of the team of Hydrology named in the Acknowledgements. LD analyzed the data and drafted the

first version of this manuscript with contributions from MW.

*Competing interests.* At least one of the (co-)authors is a member of the editorial board of Hydrology and Earth System Sciences.

### Acknowledgements

This research was conducted within the ECOSENSE project DFG SFB 1537/1, University of Freiburg. We thank the team of Hydrology, University of Freiburg, for the participation in sampler manufacturing and field work.

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
