# Peer review of "Technical note: An innovative monitoring approach to measure spatio-temporal throughfall patterns in forests"

_EGUsphere, 2025_

## Referee Comment (RC1)

Thank you to the authors for their innovative work. I hope the following review is helpful for the submitted article.

The authors have developed a novel, automated throughfall (TF) sampler that can be used to measure both throughfall and forest floor percolation. It is equipped with 3D-printed tipping buckets and a custom microcontroller board and was calibrated in the laboratory under controlled inflow. Sixty of these cost-effective samplers, each with four 600 cm² compartments, were installed by the authors in a mature, mixed temperate forest in southwestern Germany, to capture spatio-temporal variations in water percolating through the canopy and forest floor layers across different species. This monitoring setup enabled capturing of TF + forest floor interception at intra-event resolution. The sampler represents a significant methodological advancement in net precipitation measurements, and the combination of automation and network-level deployment in the study provides a valuable contribution.

The paper is well written and developed. However, I would encourage the authors to clarify few points described below and distinguish their concepts more clearly. In particular, the link between 'infiltration' (in this study, TF - forest floor interception) and 'throughfall' dynamics, as typically defined in interception studies, needs to be made clearer. Some figures and descriptions would also benefit from greater precision. Detailed comments are provided below.

**General Comments**

The monitoring network uses embedded samplers to measure water flux entering the ground. This approach is innovative, it effectively integrates both TF and forest floor percolation rather than just TF, which is typically the only factor measured in interception studies. however how its employed in this study, the collector cannot be used to calculate canopy interception loss (Eq. 1) in isolation, but rather requires support from other collectors or the exclusion of the forest floor. Or different approach to distinguish forest floor interception.

The authors do not provide an observation period that isolates TF collection alone for comparison with existing studies, which could clarify the accuracy of, or potential measurement errors in, the field setup. Further, this information can be helpful to distinguish impact of vegetation cover vs event features on TF and infiltration patterns. While the discussion focuses on TF dynamics and the influence of event characteristics, the study itself centres on estimating spatio-temporal patterns of water percolating below the surface. I would encourage the authors to reconsider the focus of the discussion and perhaps rename the instrument an infiltration sampler, since the current results cannot be directly compared with those of traditional TF studies due to the unclear separation of TF and forest floor interception. Alternatively, they could include a winter dataset (or another sampling period without foliage) to demonstrate TF-only conditions. If the sampler is the same or a follow-up to Paulsen and Weiler (2025), please discuss how forest interception and TF separation were handled in related observational work. Finally, some sentences are long and could be split for clarity (e.g., first paragraph of Section 2.1.1).

**Methods**

Figures 2 and 2.1.3 would benefit from a photo of the sampler in the field. While the authors describe the method as 'minimally invasive', digging the soil by 30 cm at multiple locations may alter the structure of the topsoil and create artificial flow paths. It is also unclear whether maintenance or troubleshooting would require re-digging each time. A photo and short explanation would greatly improve understanding.

(Line 195) It is unclear whether the collectors were positioned according to canopy density. If so, please elaborate on how many samples were placed within each canopy class (dense, medium or sparse) and the distances at which they were placed from stems in the plots.

The authors assume a two-hour interval between rainfall events based on canopy drip. However, the duration of canopy drip differs between conifers and broadleaved trees, and dense canopy cover may delay drainage further. Please elaborate on whether this interval varies between events or stands, and if so, within what range. Consider including one example event and quantifying the lag between gross precipitation and TF + forest floor infiltration measurements. As percolation through the forest floor can take even longer, a threshold of 2–4 or even 6–8 hours as is commonly used in TF studies might be more appropriate.

Also, in addition to citing Blume et al 2007 please consider to share how the C, run off coef. is adopted in this study via formula.

**Results**

The authors should consider using the same y-axis range for Figure 5. Please clarify whether the 'tipping bucket units' refer to the IDs of the individual collectors. If the collectors were placed according to canopy density, this should be indicated in the figure, as it would help to distinguish the timing of canopy storage filling from infiltration through the forest floor, the latter of which likely requires prior litter saturation.

It would also be useful to mark the start of the event and the onset of canopy drip along the x-axis.

Regarding Figure 6, are the LAI values of the sites comparable across events and stands? The difference between Events 4 and 5, which show lower variability and infiltration fractionation, may be linked to canopy storage saturation and the timing between consecutive events. Please report the time between the last event and the current event, as this could explain how antecedent conditions affect TF and infiltration patterns.

**Discussion**

(Line 330) With regard to infiltration, please comment on the time interval between events, since this directly affects canopy and forest floor interception, as well as water storage saturation.

Additionally, please consider discussing the applicability and limitations of the method in forests with dense understorey vegetation. Although the 30 cm excavation is described as 'minimally invasive', it could still disrupt the topsoil or alter flow pathways, and this should be acknowledged in the discussion.

While the discussion emphasises how event and vegetation characteristics modulate TF, the study does not include a comparison with conventional TF collectors. Therefore, it remains uncertain whether this collector can serve as an instrument suitable for estimating interception loss. The study's focus seems to be more on belowground percolation dynamics, which is an excellent contribution, but the authors should clearly decide which process (TF vs infiltration) constitutes the study's main objective.

While the event-based infiltration coefficient is a useful diagnostic metric, cross-checking it against standard throughfall collectors (e.g. funnels or troughs) at several points would be helpful. Without such validation, it is difficult to distinguish true canopy effects from sampler-specific biases (e.g. partial clogging or bypass flow).

Ultimately, however, the network and dataset make a valuable contribution to our understanding of precipitation partitioning in forests, particularly with regard to the role of vegetation structure in shaping throughfall and interception processes. If the authors wish to emphasise TF dynamics, I would encourage them to include a TF-only dataset (for example, from winter or leaf-off periods) to evaluate better intra-event TF dynamics and enable comparison with existing literature.

---

## Author Comment (AC1)

Thank you to the authors for their innovative work. I hope the following review is helpful for the submitted article.

The authors have developed a novel, automated throughfall (TF) sampler that can be used to measure both throughfall and forest floor percolation. It is equipped with 3D-printed tipping buckets and a custom microcontroller board and was calibrated in the laboratory under controlled inflow. Sixty of these cost-effective samplers, each with four 600 cm² compartments, were installed by the authors in a mature, mixed temperate forest in southwestern Germany, to capture spatio-temporal variations in water percolating through the canopy and forest floor layers across different species. This monitoring setup enabled capturing of TF + forest floor interception at intra-event resolution. The sampler represents a significant methodological advancement in net precipitation measurements, and the combination of automation and network-level deployment in the study provides a valuable contribution.

The paper is well written and developed. However, I would encourage the authors to clarify few points described below and distinguish their concepts more clearly. In particular, the link between 'infiltration' (in this study, TF - forest floor interception) and 'throughfall' dynamics, as typically defined in interception studies, needs to be made clearer. Some figures and descriptions would also benefit from greater precision. Detailed comments are provided below.

*We are happy and grateful for the positive review of the manuscript. We thank the reviewer for the valuable and helpful comments and suggest the following revisions.*

**General Comments**

The monitoring network uses embedded samplers to measure water flux entering the ground. This approach is innovative, it effectively integrates both TF and forest floor percolation rather than just TF, which is typically the only factor measured in interception studies. however how its employed in this study, the collector cannot be used to calculate canopy interception loss (Eq. 1) in isolation, but rather requires support from other collectors or the exclusion of the forest floor. Or different approach to distinguish forest floor interception.

The authors do not provide an observation period that isolates TF collection alone for comparison with existing studies, which could clarify the accuracy of, or potential measurement errors in, the field setup. Further, this information can be helpful to distinguish impact of vegetation cover vs event features on TF and infiltration patterns. While the discussion focuses on TF dynamics and the influence of event characteristics, the study itself centers on estimating spatio-temporal patterns of water percolating below the surface. I would encourage the authors to reconsider the focus of the discussion and perhaps rename the instrument an infiltration sampler, since the current results cannot be directly compared with those of traditional TF studies due to the unclear separation of TF and forest floor interception.

Alternatively, they could include a winter dataset (or another sampling period without foliage) to demonstrate TF-only conditions. If the sampler is the same or a follow-up to Paulsen and Weiler (2025), please discuss how forest interception and TF separation were handled in related observational work. Finally, some sentences are long and could be split for clarity (e.g., first paragraph of Section 2.1.1).

*We acknowledge the reviewers' comments and criticism of the limitations of the presented monitoring approach and the corresponding data particularly the direct comparison with canopy throughfall measurements. This technical note is intended to focus on introducing a novel monitoring system for interception flux components and highlight its diverse application options for measuring these components-either in isolation or in an integrated manner-across e.g. forest ecosystems of varying vegetation complexity. To strengthen the manuscripts' quality as a technical note, we propose to emphazise the novel monitoring system's design, capabilities and limitations for interception studies*

*rather than present it as a throughfall study, for which the selected dataset is too limited for this submission. Throughout the manuscript, we will direct the focus more clearly towards the strengths, constraints, and flexibility in capturing the spatio-temporal dynamics of different interception flux components isolated or integrated according to respective research objectives and ecosystem under study.*

*We acknowledge the definitions of forest interception flux components and how we refer to them were not sufficiently outlined. In the revised introduction, we will therefore provide a more differentiated description of interception fluxes in a forest ecosystem. We modify Figure 1 to explicitly mark overstory, understory and forest floor interception fluxes (different throughfall components of overstory canopy drip, understory throughfall and litter percolation) and to group these components according to their affiliation with integrated fluxes such as total vegetation interception ($I_c + I_u + I_l$), total throughfall ($TF_c + TF_u$) or litter percolation as net throughfall after interception losses ($TF_c + TF_u - I_l$) following Gerrits et al. (2010). The measurement setups presented in Figure 1 will be visually linked to these groups underlining their specific applications (e.g. with and without litter , under understory and overstory, etc.). Eq. 1 will be adapted accordingly.*

*In the Methods and Discussion sections, we will elaborate more clearly the flexible application of the sampler for measuring both isolated or integrated throughfall component, and hope to clarify potentials and limitations of presented modifications:*

*I.       Isolated measurement, e.g. overstory throughfall (canopy drip) using the sampler without litter as an alternative to traditional throughfall measurement methods; adjustable in height to target e.g. only overstory throughfall.*

*II.      Integrated measurement of net throughfall after interception losses, a new method for capturing a less frequently observed flux at high spatio-temporal resolution; additional litter trap function*

*In relation to these dual application options, we will clarify that the dataset included in this manuscript originates from an early measurement period in which leaves (unfragmented forest litter) were left inside the samplers (option II). Effectively, net throughfall (or litter percolation) was measured as an integrated flux of throughfall and forest floor interception. We selected this configuration due to its robust operation and moderate maintenance requirements. Furthermore, it provides data suited for addressing our perspective research questions on how patterns of (net) throughfall propagate into soil water content near-surface across stands of different tree species. We understand that the included data cannot be directly compared with throughfall data from isolated measurement and we will refrain from making direct linkages and interpretations without considering forest floor litter modulation.*

*Nevertheless, we believe that the example data – together with the dual application options - highlight the networks' potential to depict spatio-temporal dynamics of interception fluxes. We expect the spatial variability of throughfall from canopy drip to be sufficiently pronounced to (partially) persist through the thin, unfragmented, spatially homogenous uppermost litter layer. The spatial variability observed in integrated throughfall (litter percolations throughfall after forest floor interception) is therefore likely influenced, at least in part, by these patterns.*

*Given its flexible application, we consider our system as a throughfall sampling approach. With the here proposed revisions and clarification of throughfall component definitions, we hope to establish a consistent and precise terminology throughout the manuscript. In the Results and Discussion section, we will place particular emphasis on distinguishing isolated and integrated throughfall. We propose naming our sampler the **FluxIT-sampler** (**Flux** of **I**ntegrated **T**hroughfall **sampler**) to underline the multiple application option of the sampler and avoid further confusion about its measurement focus.*

**Methods**

Figures 2 and 2.1.3 would benefit from a photo of the sampler in the field. While the authors describe the method as 'minimally invasive', digging the soil by 30 cm at multiple locations may alter the structure of the topsoil and create artificial flow paths. It is also unclear whether maintenance or troubleshooting would require re-digging each time. A photo and short explanation would greatly improve understanding.

(Line 195) It is unclear whether the collectors were positioned according to canopy density. If so, please elaborate on how many samples were placed within each canopy class (dense, medium or sparse) and the distances at which they were placed from stems in the plots.

*We agree on the reviewers' comments and will clarify the installation and location of the samplers in the field in the final revision by a detailed overview on canopy density and stem distances at sampling locations. We will complement Figure 2 with two photos of samplers in the forest.*

*We acknowledge that installing the sampler on concrete stones and excavating an area of 25 × 25 cm inevitably introduces soil disturbance. However, this intervention is performed only once and provides a stable, level, and long-term secure base for the sampler and its tipping units. Routine maintenance thereafter requires only lifting or tilting the sampler sideways. We have also thought of other approaches in the meantime which are much less invasive and require only an area of 4 times 10cm in diameter and 20cm depth: the setup could involve PVC pipes instead of concrete stone bases. Freely swinging, automatically levelled tipping buckets are feasible too and would reduce the requirement of a precisely levelled sampler base. In addition, both container size and the height of installation are flexible and adaptable to study purpose and system to be monitored. Euro-containers are available in various dimensions, a simple stainless-steel frame cloud be used to place the samplers e.g. above ground or above understory vegetation.*

The authors assume a two-hour interval between rainfall events based on canopy drip. However, the duration of canopy drip differs between conifers and broadleaved trees, and dense canopy cover may delay drainage further. Please elaborate on whether this interval varies between events or stands, and if so, within what range. Consider including one example event and quantifying the lag between gross precipitation and TF + forest floor infiltration measurements. As percolation through the forest floor can take even longer, a threshold of 2–4 or even 6–8 hours as is commonly used in TF studies might be more appropriate.

Also, in addition to citing Blume et al 2007 please consider to share how the C, run off coef. is adopted in this study via formula.

*We suggest to modify Figure 5 as described in response of Results section. To emphazise the duration of canopy drip and forest floor percolation at the four different plots, the x-axis (+ 2 hours prior & + 6 hours past event) is extended, precipitation time series and marks for event start and end are added.*

*The example events of April 2025 included in this manuscript indicate that, at most locations, throughfall and litter percolation ceased within approximately 2 hours after the event. Occasional single tips recorded (approx. 0.05 mm each) 3 hours or later can mostly be attributed to minor additional rainfall input rather than delayed throughfall and litter percolation. Laboratory tests of percolation through unfragmented litter collected at the four field site plots support these observations, showing percolation durations of max. 1.5 hours under both dry and pre-wetted conditions.*

*Forest floor in this study refers to the upper most layer of fragmented and unfragmented organic material above the mineral soil layer (L-Of/Oh). At the field site, this layer entity is classified as a moder and approx. 10 cm thick. By litter layer (L) we refer to the loose, uppermost, unfragmented fresh litter (leaves*

*and needles) thinner than 2 cm in this case. It should be noted that only the loose, uppermost, unfragmented fresh litter is placed in the sampler. It is probable that, as litter fragments and accumulates over time, percolation durations may increase. We will add this distinction to Section 2.1.2. by explicitly differentiating between and the full forest floor profile and the here used, unfragmented litter layer.*

*With the larger dataset collected during the 2025 vegetation period, we are looking forward to investigate the above-mentioned aspects in a dedicated, separate throughfall study. We will then focus on intra-event dynamics, including canopy storage saturation time, ranges of duration of canopy drip and litter percolation across different tree species, events and antecedent wetness conditions. We will also test different rainfall and throughfall event segmentation thresholds (2; 4; 6; 8 hours) to assess their influence on the sampled water amounts. The formula for $C_i$ will be added to this section together with a detailed explanation of how it was adopted from the runoff coefficient.*

**Results**

The authors should consider using the same y-axis range for Figure 5. Please clarify whether the 'tipping bucket units' refer to the IDs of the individual collectors. If the collectors were placed according to canopy density, this should be indicated in the figure, as it would help to distinguish the timing of canopy storage filling from infiltration through the forest floor, the latter of which likely requires prior litter saturation.

It would also be useful to mark the start of the event and the onset of canopy drip along the x-axis.

Regarding Figure 6, are the LAI values of the sites comparable across events and stands? The difference between Events 4 and 5, which show lower variability and infiltration fractionation, may be linked to canopy storage saturation and the timing between consecutive events. Please report the time between the last event and the current event, as this could explain how antecedent conditions affect TF and infiltration patterns.

*We fully agree with the reviewers' suggestions. By adding precipitation time series to Figure 5 & 6 and a regularly spaced x-axis (time in days) to Figure 6, the intra-event timings and the influence of antecedent wetting conditions will be presented more clearly.*

*LAI values obtained from hemispheric photographs for April and May 2025 (one date before and one date after the events 1-6; data provided by Chair of Remote Sensing, University of Freiburg) vary slightly between the stands, tree species and events (10-day period after foliation developed). They will be added to Figure 6. In a forthcoming throughfall study based on the full 2025 vegetation period, we aim to investigate in detail the above-named influences of changing LAI values, variable precipitation intensities and duration as well as pre-wetted conditions on canopy storage (saturation) and the generation of throughfall patterns.*

**Discussion**

(Line 330) With regard to infiltration, please comment on the time interval between events, since this directly affects canopy and forest floor interception, as well as water storage saturation.

*see response Results section*

Additionally, please consider discussing the applicability and limitations of the method in forests with dense understorey vegetation. Although the 30 cm excavation is described as 'minimally invasive', it could still disrupt the topsoil or alter flow pathways, and this should be acknowledged in the discussion.

While the discussion emphasises how event and vegetation characteristics modulate TF, the study does not include a comparison with conventional TF collectors. Therefore, it remains uncertain whether this

collector can serve as an instrument suitable for estimating interception loss. The study's focus seems to be more on belowground percolation dynamics, which is an excellent contribution, but the authors should clearly decide which process (TF vs infiltration) constitutes the study's main objective.

*We agree with the remarks and will refocus the discussion on the applicability and limitations of the monitoring network for measuring different interception fluxes and debate perspective modifications to the setup a.o. for forests with dense understory vegetation. The discussed methodological advantages of the system compared to traditional throughfall measurements remain valid as method 1 represents a direct alternative to discussed throughfall measurement methods. For less invasive application approaches please refer to response in Methods section.*

While the event-based infiltration coefficient is a useful diagnostic metric, cross-checking it against standard throughfall collectors (e.g. funnels or troughs) at several points would be helpful. Without such validation, it is difficult to distinguish true canopy effects from sampler-specific biases (e.g. partial clogging or bypass flow).

*We agree. As explained in detail above, we will focus the manuscript on the sampling approach and its flexible application. Using the recently collected data, we look forward to investigating the spatio-temporal characteristics of throughfall as part of a dedicated throughfall study.*

Ultimately, however, the network and dataset make a valuable contribution to our understanding of precipitation partitioning in forests, particularly with regard to the role of vegetation structure in shaping throughfall and interception processes. If the authors wish to emphasise TF dynamics, I would encourage them to include a TF-only dataset (for example, from winter or leaf-off periods) to evaluate better intra-event TF dynamics and enable comparison with existing literature.

---

## Author Comment (AC2)

This paper proposed monitoring system for throughfall with tipping bucket instrument, which authors had developed recently (Paulsen and Weiler, 2025). I agree that spatial distribution of throughfall is highly heterogeneous, and interception studies require instruments which can measure the spatial distribution of throughfall easily and validly. So, I could understand authors' motivations shown in this paper, but I regret to say that this paper should be rewritten very carefully.

*We thank the reviewer for the valuable and helpful comments and for reviewing our manuscript. We acknowledge the criticism and suggest the following revisions.*

When tipping bucket system is applied to measure inflow of water, static and dynamic calibrations are necessary, as authors cited Shimizu et al. (2018). Authors did very careful calibration of static volume of one tip. Also, increasing underestimation of one tip for increasing inflow must be considered. The static one tip volume is 2.21 to 3.69 mL, ranging 0.037 mm to 0.062 mm, relatively high resolution for gross rainfall and throughfall measurement. Please note that underestimation occurs at every tip under high intensity, and number of tips should increase with higher resolution (smaller volume of one tip), leading to larger underestimation compared with tipping bucket of low resolution (larger volume). As authors stated, throughfall sometimes exceeds gross rainfall, strongly requiring applying correction based on the dynamic calibration. This is "technical note", in which well-known uncertainties must be evaluated carefully. The one tip volume is similar with Onset and Davis rain gauges, and their dynamic calibration curve had been described in Iida et al. (2012, Hydrological Processes) and Iida et al. (2018, Journal of Hydrometeorology).

*We agree with the reviewers' comment and acknowledge that measurements of water amounts based on the tipping bucket principle may lead to bias under dynamic conditions, depending on the inflow intensity into the tipping bucket. In our case, the sampler design partially mitigates this issue. The pipes connecting the drainage compartments to the tipping bucket units have an inner diameter of 3 mm, which limits the inflow rate. During high-intensity rainfalls incoming water is collected in the sampler, funnelled and temporally stored in the four drainage compartments that function as a buffer emptying at 0.4 L min$^{-1}$. We will describe this feature of the sampler design more precisely in section 2.1.3. In general, our measurement approach is designed to appropriately capture the spatio-temporal patterns of most events; extreme events are not the primary focus.*

*We appreciate the reviewers' suggestion to consider dynamic calibration. As described in section 3.1., we calibrated all single 240 tipping bucket units individually using static calibration at an already relative high rainfall intensity of 40 mm h$^{-1}$, corresponding to 13.94 tips minute$^{-1}$. To evaluate under- and overestimation for lower and higher intensities, we conducted a dynamic calibration on a subset of the tipping bucket units (n = 10). At highest tested rainfall intensity of 120 mm h$^{-1}$, in average 27.83 tips per 100 mL were recorded. Using the mean tip volume from static calibration (2.87 mL), this indicates an undercatch of approximately 23% at very high intensity rainfalls. During 1.5 years of observation, rainfall intensities at the field site exceeded 120 mm h$^{-1}$ for 11 minutes (0.015% of total rainfall duration in 1.5 years) and 40 mm h$^{-1}$ for 2.4 hours (0.2% of total rain duration). The resulting total underestimation from extreme events is less then 0.05%, which we consider acceptable . Like any method, tipping bucket measurements have their limitations. Still, they are simple in principle and cost-effective allowing us to realise an approach for fine-scale spatio-temporal monitoring.*

*We would also like to emphazise the advantage of the flexible dimensioning of all components of this self-built sampler-including the drainage compartments volume and size, the pipe diameter and tipping bucket volumes made possible through 3D-printing and the use of modular Euro-containers. The selected dimensions of this setup were selected to match the prevailing and most frequent rainfall intensity ranges at the study site. Should field observations and data analysis require rainfalls intensities exceeding the inflow capacity, the user can plan to exchange the measurement units with larger components. This scalability supports also the application of this monitoring approach to drier or wetter climates.*

*Moreover, we will analyse the recently collected, larger dataset from the 2025 vegetation period with regard to over- and underestimation of the tipping bucket units. We will evaluate rainfall events of varying intensities and test correction approaches for measurements during high intensity rainfall following e.g. Iida et al. (2012, Hydrological Processes) and Iida et al. (2018, Journal of Hydrometeorology).*

Title of this paper is "... to measure spatio-temporal throughfall patterns in forests". However, reading the current manuscript, main results are rainwater passing through the litter manually put on the capturing area. Changing logic through the manuscript is not suitable for scientific journals. Please confirm that the current objectives (line 111-114) include throughfall only. If authors want to focus on rainwater passing through the litter, careful revisions must be required for whole manuscript.

I feel strange for expression of "infiltration" measured by this equipment. The tipping bucket measures inflow of rainwater, which passed the litter put on the capturing area of throughfall. I could not think that the inflow is the same as infiltration into soil. To declare infiltration, at least, initial soil water condition, inflow intensity, infiltration capacity and topography (slope or flat) should be considered.

*We understand the comment and acknowledge the definitions of forest interception flux components and how we refer to them were not sufficiently outlined. As also explained in the answer to the second reviewer of this manuscript, we will therefore provide a more differentiated description of interception fluxes in a forest ecosystem. In addition, we propose to strengthen the manuscripts' quality as a technical note introducing a novel monitoring system for interception flux components rather than present it as a throughfall study, for which the dataset was too limited at the moment of submission. We hope to thereby clarify the objective of this manuscript.*

*In the revised introduction, we will therefore provide a more differentiated description of interception fluxes of forest ecosystems. We modify Figure 1 to explicitly mark overstory, understory and forest floor interception fluxes (different throughfall components of overstory canopy drip, understory throughfall and litter percolation) and to group these components according to their affiliation with integrated fluxes such as total vegetation interception ($I_c + I_u + I_l$), total throughfall ($TF_c + TF_u$) or litter percolation as net throughfall after interception losses ($TF_c + TF_u - I_l$) following Gerrits et al. (2010). The measurement setups presented in Figure 1 will be visually linked to these groups underlining their specific applications (e.g. with and without litter, under understory and overstory, only overstory, etc.). Eq. 1 will be adapted accordingly.*

*In the Methods and Discussion sections, we will elaborate more clearly the flexible application of the sampler for measuring both isolated or integrated throughfall component, and hope to clarify potentials and limitations of presented modifications:*

*I.        Isolated measurement, e.g. overstory throughfall (canopy drip) using the sampler without litter as an alternative to traditional throughfall measurement methods; adjustable in height to target e.g. only overstory throughfall.*

*II.        Integrated measurement of net throughfall after interception losses, a new method for capturing a less frequently observed flux at high spatio-temporal resolution; additional litter trap function*

*Given its flexible application, we consider our system as a throughfall sampling approach. With the here proposed revisions and clarification of throughfall component definitions, we hope to establish a consistent and precise terminology throughout the manuscript. In the Results and Discussion section, we will place particular emphasis on distinguishing isolated and integrated throughfall. We propose naming our sampler the **FluxIT-sampler** (**Flux** of **I**ntegrated **T**hroughfall **sampler**) to underline the multiple application option of the sampler and avoid further confusion about its measurement focus.*

[Specific comments]

Line 7-21 (abstract)

There is no "infiltration" among these lines. Readers could not expect that this paper describes inflow of rainwater passed through the litter put on the capturing area.

*Agreed. As stated above, we will consistently use the terms of litter percolation or integrated throughfall instead of infiltration. Furthermore, we will strengthen the focus of the abstract in correspondence to the refocusing of the overall manuscript: a technical note introducing a novel monitoring system for interception flux components and highlighting its diverse application options for measuring different interception flux components —either in isolation or in an integrated manner—across e.g. forest ecosystems of varying vegetation complexity.*

Line 25-85

Detailed background for interception process including throughfall measurement is written here, but these topics are not investigated in this manuscript. I agree that sample size is important issue for throughfall studies, but I could not understand how this equipment contributes to this issue.

*By providing a detailed background on interception components in a forest ecosystem, we underline the complexity of the overall process and the connectivity of individual components, such as overstory and understory throughfall. As described above, we will strengthen the link of interception component description and corresponding monitoring approaches.*

*Regarding the sample size and its relevance in the proposed monitoring scheme, our setup includes 240 tipping units distributed across the field enabling the capture of spatio-temporal dynamics of both integrated or isolated interception fluxes- a feature rarely achieved by other interception measurement methods. For the example of integrated throughfall measurements, we expect the spatial variability of throughfall from canopy drip across a plot to be sufficiently large to (partially) persist through the unfragmented, uppermost litter layer. Consequently, the spatial variability of integrated throughfall (litter percolations throughfall after forest floor interception) measured with our samplers is a result or at least influenced by these patterns. This effect is particularly evident because the unfragmented litter layer in the samplers is thin and relatively homogeneous within each plot, limiting any additional homogenization or heterogenization of throughfall patterns.*

Line 192-212

It is somewhat difficult for me to understand the situation. Is it correct that newly developed instruments here were used for only measurement of rainwater inflow passing litter? Troughs collect throughfall, but are these data shown in this MS? If trough data is not used, related sentences must be removed. Rainwater captured by trough was measured by Davis rain gauge, so actual one tip should be very small. Please note that underestimation by Davis gauge is high (Iida et al., 2018), and careful attention should be paid for measurements.

*As described in detail above we will address in larger detail the flexible application of the sampler to measure different throughfall components in an isolated or integrated manner and hope to clarify potentials and limitations of presented modifications. But we would exclude the description of traditional trough throughfall measurements to avoid confusion.*

Line 245-294, 295-351

These results are "infiltration", that is rainwater inflow passed the litter. However, in discussion section, results are compared with previous throughfall studies. Readers should feel strange. In my

opinion, as this equipment is throughfall measurement system, authors should compare the throughfall measurements by this new system and the ordinary gauges like storage type point gauges. That is very simple and clear way to show the validity of new system.

*We will refocus the Discussion section in accordance with the reviewers' suggestions and our proposed alterations. As also expressed in our response to the second review, in the context of the dual application options, we will point out more precisely that the data included in this manuscript originate from an early measurement period in which leaves (unfragmented forest litter) were left in the samplers (option II). Effectively, in this configuration net throughfall (or litter percolation) was measured as an integrated flux of throughfall and forest floor interception. We understand that for comparison of the included data to throughfall data from traditional measurement methods, we need to consider pattern modulation by the forest floor litter. However, we believe the example data in combination with the dual application options highlight the networks' potential to depict spatio-temporal dynamics of interception fluxes.*

*With the larger dataset collected during the 2025 vegetation period, we look forward to investigating the above-mentioned aspects in a detailed throughfall study. We also consider a measurement period using empty samplers (no litter) to collect data directly comparable with traditional throughfall studies.*

Line 353-357

These are well-known uncertainties for tipping bucket, which should be considered carefully in this manuscript.

*Agreed, we will consider this aspect more strongly in the manuscript. Please refer to the detailed answer on over-/underestimation and dynamic calibration above*

Line 370

Not all readers know ECOSENSE. Please take care of readers.

*Agreed, we will add a brief description of the ECOSENSE project and include a citation of the publication presenting the field site (Tesch et al. 2025 https://doi.org/10.5194/egusphere-2025-4979).*

[Technical corrections]

Line 166

Please check [ tips" ].

*Agreed*

**Citation**: https://doi.org/10.5194/egusphere-2025-4285-RC2